# Curriculum Learning for Language-guided, Multi-modal Detection of Various Pathologies

**Laurenz Adrian Heidrich**[*1,2]        LAURENZ.HEIDRICH@TUM.DE

**Aditya Rastogi**[*2]        ADITYA.RASTOGI@UKBONN.DE

**Priyank Upadhya**[2]        PRIYANK.UPADHYA@UKBONN.DE

**Gianluca Brugnara**[2,3]        GIANLUCA.BRUGNARA@UKBONN.DE

**Martha Foltyn-Dumitru**[2]        MARTHA.FOLTYN-DUMITRU@UKBONN.DE

**Benedikt Wiestler**[1,4]        B.WIESTLER@TUM.DE

**Philipp Vollmuth**[2,3]        PHILIPP.VOLLMUTH@UKBONN.DE

[1] *AI for Image-Guided Diagnosis and Therapy (AI-IDT), School of Medicine and Health, Technical University of Munich, Munich, Germany*

[2] *Division for Computational Radiology & Clinical AI (CCIBonn.ai), University Hospital Bonn, Bonn, Germany*

[3] *Divison for Medical Image Computing (MIC), German Cancer Research Center (DFKZ), Heidelberg, Germany*

[4] *Munich Center for Machine Learning, Munich, Germany*

**Editors:** Accepted for publication at MIDL 2025

## Abstract

Pathology detection in medical imaging is crucial for radiologists, yet current approaches that train specialized models for each region of interest often lack efficiency and robustness. Furthermore, the scarcity of annotated medical data, particularly for diverse phenotypes, poses significant challenges in achieving generalizability. To address these challenges, we present a novel language-guided object detection pipeline that leverages curriculum learning strategies, chosen for their ability to progressively train models on increasingly complex samples, thereby improving generalization across pathologies, phenotypes, and modalities. We developed a unified pipeline to convert segmentation datasets into bounding box annotations, and applied two curriculum learning approaches - teacher curriculum and bounding box size curriculum - to train a Grounding DINO model. Our method was evaluated on different tumor types in MRI and CT scans and showed significant improvements in detection accuracy. The teacher and bounding box size curriculum learning approaches yielded a 4.9% AP and 5.2% AP increase over baseline, respectively. The results highlight the potential of curriculum learning to optimize medical image analysis and clinical workflow. The code is available at https://github.com/CCI-Bonn/CL4OD.

**Keywords:** Medical Image Analysis, Deep Learning, Tumor Detection, Curriculum Learning

## 1. Introduction and Related Work

Pathology detection on medical imaging is a cornerstone of modern radiology practice and plays a crucial role in diagnosis and treatment planning. Accurate pathology detection is

---

[*] Contributed equally

critical for determining the presence, location, and extent of abnormalities, guiding diagnostic accuracy, enabling targeted treatments, monitoring disease progression, and evaluating therapy efficacy. Simplifying associated workflows and reducing prediction complexity is essential to increase efficiency, remove barriers to clinical adoption, and improve prediction quality. This can be achieved by unifying specialized models into foundation models, and by reducing algorithmic and task complexity. In this context, segmentation algorithms can be substituted by detection-only algorithms for many clinical tasks, such as identifying new metastases or counting the number of existing lesions.

Recent advances in medical imaging have led to the development of foundation models capable of handling multiple modalities and interactive tasks (Ma et al., 2024a,b). These models show superior flexibility compared to specialized segmentation networks such as nnU-Net (Isensee et al., 2021), operating on different imaging modalities and accepting visual input prompts. In addition, recent developments have also demonstrated the benefits of language guidance in medical image analysis, enabling more efficient medical image interpretation, with several studies focusing on language-driven segmentation (Koleilat et al., 2024b; Li et al., 2024; Liu et al., 2023; Zhao et al., 2025) and improving zero-shot and few-shot performance (Koleilat et al., 2024a). Notable work has been done in the area of medical object detection, particularly for brain tumors (Mercaldo et al., 2023; Chen et al., 2024a; Abdusalomov et al., 2023; He et al., 2023). However, there appears to be a lack of multi-modal and multi-pathology detection frameworks. The success of Grounding DINO (G-DINO) (Liu et al., 2024), an open-set language-guided object detector, has generated interest in its application to medical imaging, allowing the integration of different pathologies, modalities and text prompts into a single network. So far, only a few algorithms (Biswas, 2023; Xie et al., 2024; Ramesh et al., 2023) have taken advantage of this additional guidance for object detection in medical imaging. Additionally, these works do not investigate G-DINO in detail, as a stand-alone architecture, but rather use it as a box prompt generator for SAM, following the idea of Grounded SAM (Ren et al., 2024). In this study we focus on developing a language-guided network to detect pathologies - specifically tumors - of various organs.

Unlike natural images, tumor detection poses unique challenges due to the significant variability in tumor phenotypes across patients, which demands large datasets to achieve generalization. However, the scarcity of annotated medical imaging data makes generalization challenging, especially for detecting smaller tumors (Abdusalomov et al., 2023; He et al., 2023). Foundation segmentation models such as MedSAM (Ma et al., 2024a) even ignore this issue by entirely excluding pathologies with a volume less than 1000 pixels and a cross-sectional area less than 100 pixels. To address these shortcomings, we investigated different Curriculum Learning (CL) strategies (Bengio et al., 2009) to increase detection accuracy. CL, introduced by Bengio et al. (Bengio et al., 2009), has found several applications in the medical domain (Jiménez-Sánchez et al., 2019; Wei et al., 2021; Oksuz et al., 2019; Fischer et al., 2024) with the goal of improving performance by gradually increasing training complexity. The strategy used in this study, called data-level CL, gradually increases the complexity of the training samples: First, the model is trained on large, well-contrasted tumors to establish robust feature representations. Next, the training data is expanded to include smaller, less conspicuous tumors with increasing anatomical and modality vari-

ability. This progressive learning approach helps the model develop better generalization capabilities and improves its sensitivity to subtle pathological findings.

In this work, we explore the potential of two different CL strategies on G-DINO's detection performance by pre-training the network on the TotalSegmentator dataset (Wasserthal et al., 2023), followed by CL-based fine-tuning on tumor datasets spanning different imaging modalities (Magnetic Resonance Imaging (MRI) and Computed Tomography (CT)) and anatomical sites (brain metastasis, glioma, liver & kidney tumor). In addition, we have developed a pipeline to convert ground truth segmentations into bounding boxes by using morphological operations to consolidate them. This goes beyond the naive approach of drawing tight bounding boxes around segmentations. An extensive evaluation of the G-DINO baseline model was conducted, comparing its performance with models trained using two CL approaches: teacher CL (Weinshall et al., 2018) and bounding box CL (Shi and Ferrari, 2016). The results show a 4.9% improvement in Average Precision (AP) with teacher CL and a 5.2% increase in AP with bounding box CL compared to the baseline model. Based on the reviewed literature, this paper is among the initial efforts to:

1. Apply two different CL strategies to a language-guided detection network (G-DINO).
2. Train G-DINO jointly on different pathologies from various body regions and modalities, demonstrating the model's versatility with limited datasets.
3. Develop and formalize a novel preprocessing pipeline to convert medical segmentation datasets into object detection datasets.

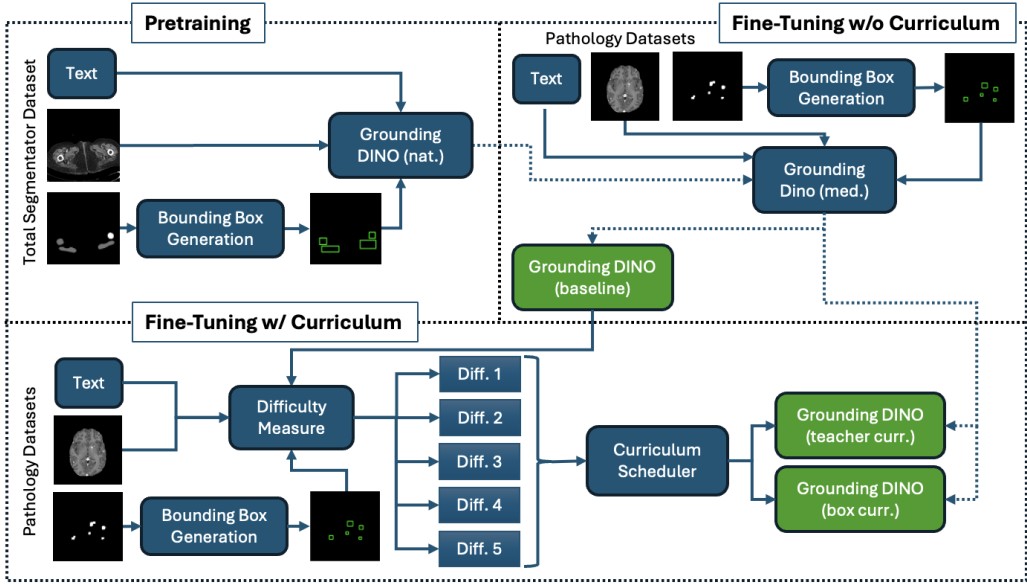

Figure 1: Overview of our method: As a first step, the natural image G-DINO model is pre-trained on the Total Segmentator dataset (top left). In a second step, the baseline is finetuned without CL on all pathology datasets (top right). Finally, two CL models are trained: for teacher CL, the baseline is used to guide the difficulty sorting, while for bounding box CL, the size of bounding boxes are used for difficulty sorting (bottom).

## 2. Method

Our methodological pipeline consists of three parts, as shown in Figure 1. The first part consists of pre-training the language-guided detection network G-DINO on a large multi-modal, multi-organ dataset to detect 163 different regions of interest - 104 for CT and 59 for MRI. The second part consists of fine-tuning the pre-trained G-DINO on the pathological datasets producing the baseline. Finally, the third part consists of fine-tuning the pre-trained model on the pathological datasets using two different CL strategies.

### 2.1. Ground Truth Bounding Box Generation

Open-source medical object detection datasets are even more scarce than segmentation datasets. To leverage the relative abundance of medical segmentation datasets, we devised an efficient method to generate ground truth bounding boxes from existing segmentation data. To generate bounding boxes for the pathology datasets, we first removed all segmentation masks not related to pathologies, such as liver or liver vessel masks. Then, we merged masks if certain tumors had compartments (e.g., "contrast enhancing" and "necrotic" parts). Finally, we performed dilation on the binary segmentation mask to remove discontinuities in the tumor mask. This provides a more accurate bounding box for a tumor instead of separate bounding boxes for discontinuous regions of the same tumor and ensures that the bounding boxes are not overly tight, providing a more realistic representation similar to human annotation. We perform the dilation in 2 iterations with a $3 \times 3$ kernel. After performing the dilation, we then drew tight bounding boxes around the resulting segmentations (see Figure 2). Models like MedSAM, which use oracle bounding boxes as training prompts, address noisy boxes by discarding segmentation masks below a size threshold. Our approach mitigates noise while retaining masks for small tumors.

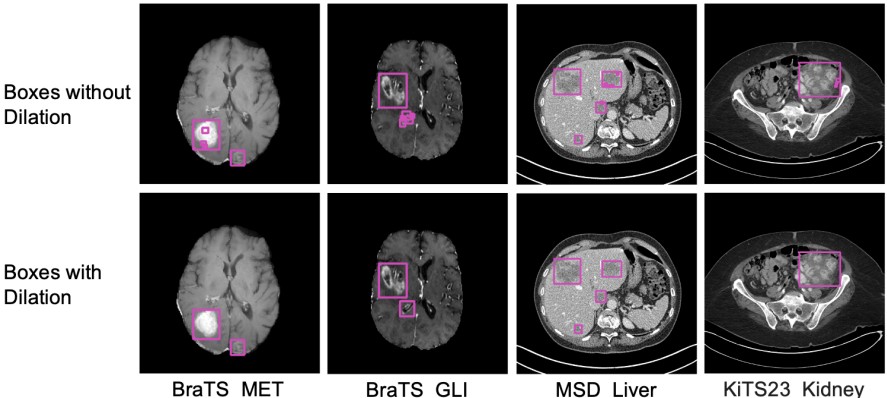

Figure 2: Depiction of the regularization effects of our bounding box pipeline using dilation.

### 2.2. Grounding DINO

G-DINO (Liu et al., 2024) is an open object detector capable of identifying any object based on textual input, such as referring expressions or categories. Given an (Image, Text) pair input, G-DINO predicts multiple (Bounding Box, Noun Phrase) pairs with confidence

scores for each detected entity. The noun phrase is the predicted semantic entity of the box and is derived from the input prompt in an open-set fashion. The model employs a dual-encoder-single-decoder architecture consisting of an image encoder for visual feature extraction, a text encoder for textual information processing, a feature enhancer for fusion of extracted features, a language-guided query selection module for query initialization, and a cross-modality decoder for bounding box refinement (Liu et al., 2024). We used the G-DINO implementation from the mmdetection framework (Zhao et al., 2024) and adopted the focal loss (Lin et al., 2020) ($\gamma = 2.0$ and $\alpha = 0.25$) and the weighted L1 loss ($w = 5$) as loss functions. (Zhao et al., 2024) For image and text encoders, we used Swin-Tiny (Liu et al., 2021) and bert-base-uncased (Devlin et al., 2019), respectively. The text prompt is constructed by concatenating all possible class names. Thus, the fine-tuning prompt for all training images was "glioma . brain_metastasis . liver_tumor . kidney_tumor".

## 2.3. Curriculum Learning

Weinshall et al. (Weinshall et al., 2018) theoretically showed that in linear regression, convergence decreases with increasing sample difficulty. Empirically, they found that in non-convex optimization, higher difficulty increases gradient variance, slowing convergence and worsening generalization compared to CL. Based on this, we propose two difficulty-sorting methods: teacher CL and bounding box CL. For the teacher CL, we used the baseline model to perform inference on the entire training dataset of pathological images and computed the AP score for each image to sort them into five difficulty levels based on their evaluation scores (ranging from 1 for the easiest to 5 for the most difficult). The baseline model acts as a difficulty grader, assuming high-precision samples are easier to learn, while low-precision ones are more challenging. For false positives, we manually set the AP score to 0.0 if the network predicts a bounding box with a confidence level greater than 0.3. In bounding box-based curriculum, difficulty is defined by bounding box size, as multiple studies (including original G-DINO) show precision scores increase with larger boxes. The bounding box curriculum classifies each training sample into one of five difficulty levels based on the size of the smallest bounding box present in the image. We found that bounding box size is a primary indicator of prediction difficulty, as shown by the performance of the baseline model (Table 1). This approach has the advantage of not requiring a trained baseline model a priori. In both CL approaches, we randomly assigned training samples without bounding boxes (i.e., no object of interest) to difficulty categories to maintain an equal distribution. To fine-tune the model on the pathological dataset using CL, we start with the easiest category and progressively introduce more difficult categories at each epoch. After 5 epochs, the network is fine-tuned on the complete data until convergence.

## 3. Experiments and Results

### 3.1. Pretraining & Datasets

In our experiments, we initialized the G-DINO architecture with weights published by (Zhao et al., 2024), which were trained on several natural image datasets (Objects365 (Shao et al., 2019), GRIT (Peng et al., 2023), V3Det (Wang et al., 2023), Golden-G dataset (Kamath et al., 2021)). We first pretrained this network on the TotalSegmentator dataset,

which consists of CT (Wasserthal et al., 2023) and MRI (Akinci D'Antonoli et al., 2024) scans of the entire human body, to adapt the image encoder weights to medical imaging modalities and to improve medical semantic understanding. We then fine-tuned the network on heterogeneous datasets spanning multiple modalities, pathologies, hospitals, and scanner manufacturers. This aggregated dataset includes both MRI and CT scans with four different detection targets: brain metastasis, glioma, liver & kidney tumor. A detailed overview of all datasets used in this work is given in Table A1 in the Appendix.

### 3.2. Data Preprocessing & Training Detail

Both CT and MRI datasets, originally in 3D NIfTI format, require preprocessing for compatibility with G-DINO, a 2D object detector. Following Ma et al. (Ma et al., 2024a), we clipped MRI images to their $[0.05, 99.5]$ percentile and normalized them to $[0, 255]$, while CT images were windowed (level $= 40$ and width $= 400$) before normalization. For the pathological datasets, we used organ segmentation masks to only retain slices containing the organ of the associated pathology. Patients from all datasets were split into train/validation/test $(0.7, 0.15, 0.15)$ sets. The resulting 2D training dataset consisted of 199,672 slices, of which 66,990 slices had bounding box annotations (i.e., tumors were present). During training, the BraTS_Glioma dataset was undersampled by a factor of 3 to ensure a balanced class distribution. We trained the models on two NVIDIA RTX A6000 GPUs with a batch size of 10 until convergence. The baseline model was trained with a learning rate of $1e^{-5}$ for the first 5 epochs, followed by a learning rate of $1e^{-6}$, while the curricula models were trained with $1e^{-5}$ for the first 7 epochs and $1e^{-6}$ for the remaining epochs to ensure equal data exposure and to compensate for shorter curricula epochs due to data exclusion. Figure A1 in the appendix illustrates the training loss and validation curves.

### 3.3. Curriculum Learning - Difficulty categories

We categorized the training data into five difficulty levels based on two heuristics to obtain two CL strategies as explained in Section 2.3. After training the baseline model, we observed that the size of the bounding box correlated with the precision score and thus could be an indicator of the difficulty of the samples (Table 1). To standardize bounding box sizes, we calculated their areas relative to the image area. We then sorted the samples based on the smallest bounding box present in each slice, ensuring an even distribution across categories. Slices without bounding boxes were randomly assigned to maintain an equal distribution of annotated and unannotated samples. For teacher CL, we additionally included slices without ground-truth bounding boxes but with false positive predictions from the baseline model and assigned them an AP score of 0.0. We then created intervals to maintain a roughly equal distribution across categories. Dataset distributions across categories are shown in the appendix for both heuristics in Table A2 and Table A3, respectively.

### 3.4. Experimental Setup

After pre-training on the TotalSegmentator dataset, we simultaneously fine-tuned G-DINO on the full pathological training dataset to obtain three models: a baseline model and two CL-based models using the bounding box and teacher principles introduced in Section 2.3. For the baseline, the training data was sampled entirely randomly. All weights were updated

| Dataset | Curr. | AP (%) @0.5 | AP (%) @0.75 | AP (%) | AP (%) large | AP (%) medium | AP (%) small |
|---|---|---|---|---|---|---|---|
| Overall | Without | 69.7 | 50.6 | 46.5 | 69.3 | 62.0 | 30.9 |
| | Box | **75.7** | **56.7** | **51.7** | 70.0 | **65.8** | **35.6** |
| | Teacher | 75.5 | 56.0 | 51.4 | **72.7** | 65.5 | **35.6** |
| Yale_BM | Without | 61.5 | 48.2 | 42.2 | - | 69.9 | 37.3 |
| | Box | **79.5** | **64.9** | **56.9** | - | **81.5** | **52.4** |
| | Teacher | 76.0 | 61.8 | 54.3 | - | 77.5 | 50.8 |
| BraTS_MET | Without | 66.7 | 54.1 | 45.9 | - | 76.9 | 42.2 |
| | Box | **82.7** | **65.9** | **57.4** | - | **83.8** | **54.2** |
| | Teacher | 81.5 | 64.5 | 56.6 | - | 83.6 | 53.0 |
| BraTS_GLI | Without | 84.8 | 72.2 | 65.8 | - | 81.1 | 48.1 |
| | Box | **85.2** | **72.5** | **66.0** | - | **81.3** | **48.4** |
| | Teacher | 84.5 | 71.8 | 65.4 | - | 80.7 | 47.5 |
| MSD_Liver | Without | **61.4** | 32.1 | **33.2** | **66.6** | **46.8** | 19.9 |
| | Box | 60.4 | **33.2** | 32.7 | 63.1 | 44.9 | **21.9** |
| | Teacher | 61.3 | 30.2 | 31.9 | 65.8 | 46.0 | 19.0 |
| MSD_Hep_Vessel | Without | 69.4 | 37.7 | 39.2 | 62.8 | 43.7 | 15.0 |
| | Box | 70.5 | **43.5** | **43.1** | 67.8 | **47.5** | 17.6 |
| | Teacher | **72.4** | 42.9 | 42.9 | **68.5** | 46.4 | **17.8** |
| KiTS23_Kidney | Without | 74.6 | 59.5 | 52.9 | 78.4 | 53.8 | 22.6 |
| | Box | 75.7 | 60.4 | 53.9 | 79.0 | 56.0 | 19.0 |
| | Teacher | **77.0** | **64.5** | **57.4** | **83.8** | **58.6** | **25.2** |

Table 1: Overview of the AP scores on individual datasets and averaged to an overall score.

during training, including the image and text encoders. During training, model weights were evaluated on a merge of the individual validation sets, and the best performing (mean-AP over all detection targets) weights were selected. Subsequent testing was performed on all datasets individually. We evaluated the object detection results using the COCO metrics (Lin et al., 2014). We used AP[1] values at different IoU thresholds: AP@0.5 and AP@0.75 with IoU thresholds of 0.5 and 0.75, respectively. The unspecified AP represents the average metric across IoU thresholds between 0.5 and 0.95 in 0.05 increments. Additionally, we evaluated the predictions separately for different bounding box sizes, where AP small, AP medium, AP large refer to the AP of ground truth bounding boxes with areas of $[0, 32^2]$, $[32^2, 96^2]$, and $[96^2, \infty)$ pixels, respectively.

### 3.5. Results

Pretraining takes around 4 days while fine-tuning takes around 2.5 days for each model. Teacher CL additionally requires 2.5 days of baseline training beforehand and approximately 5 hours for evaluating the baseline on the entire training set to assign difficulty classes to each sample. Table 1 shows the quantitative test results for all models. Figure 3 shows the predictions of our three models alongside the ground truth for one case from each dataset. Both CL approaches improved performance on average AP metrics, with the bounding box

---

1. Standard deviation cannot be computed for single-class AP scores from a single model, as AP is a single summary value — the area under the precision-recall curve — rather than a distribution.

CL model achieving the highest gains (+5.2% AP, +6.1% AP@0.75, +6.0% AP@0.5 over baseline). Both CL models outperformed the baseline in all size-constrained AP scores, with the largest gains in the most difficult categories (AP small: +4.7% for both models, AP medium: +3.5% & +3.8% for bounding box CL and teacher CL, respectively). The results thus support our hypothesis that CL improves performance especially for the most difficult samples with the smallest tumors. Looking at individual datasets, the CL models performed best in 5 out of 6 datasets for all metrics, with the MSD_Liver dataset showing slight underperformance in this context (-0.5% AP for bounding box CL, -1.3% AP for teacher CL). Overall, the results indicate that CL generally improves model performance, especially for challenging detection tasks with small to medium bounding boxes. Figure A2 in the Appendix shows the density distribution across categories as the teacher model trains. After training, the distribution shifts toward the "Easiest" category.

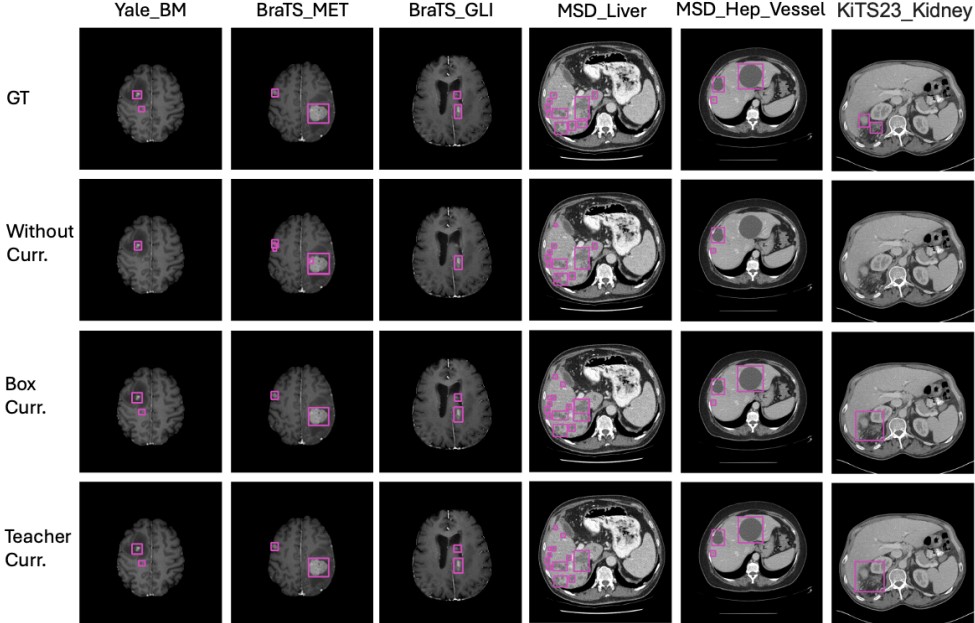

Figure 3: Visualization of the predicted results with an illustrative case from each dataset.

### 3.6. Ablation Studies

**Anti-Curriculum**: To show the effect of CL, and also to follow recent work (Wu et al., 2023; Chen et al., 2024b; Braun et al., 2017) that proposes a hard-to-easy methodology (anti-CL), we also trained our models in such a setting. The results are shown in the appendix in Table A4. Anti-box CL achieves an overall AP score of 49.5%, which is 3.0% better than the baseline and 2.2% less than regular bounding box CL. Anti-teacher CL, on the other hand, achieves an overall AP score of 51.2%, which is 3.7% better than the baseline and only slightly (0.2%) worse than regular teacher CL. Both CL and anti-CL proved to be effective for both difficulty sorting approaches and outperformed the baseline. Our results show that when training samples were sorted by difficulty based on the performance of the baseline model, the sorting order had little effect on final accuracy. However, when

the difficulty sorting was based on bounding box size, the sample order during training had a more pronounced effect, leading to greater variation in accuracy. This suggests that difficulty sorting based on manual heuristics interacts more with learning dynamics than teacher-based difficulty sorting.

**Finetuning Modality:** In this experiment we try to determine the individual contributions of each modality and the overall benefit of their combination during finetuning. Table A5 in the Appendix G presents test scores for two models, each finetuned on a single modality without CL. The results do not indicate a clear advantage of fine-tuning on a single modality versus multiple modalities. As expected, the CT model performs poorly on MRI datasets and vice versa. When tested on the same modality, its performance is comparable to multi-modal networks Table 1. Specifically, the CT model underperforms compared to the best model across all three CT datasets, while the MRI model achieves performance similar to the multi-modal fine-tuned CL algorithm.

**Pretraining:** To evaluate the effect of pretraining, we perform multiple ablations. Firstly, we test the natural image & medically pretrained G-DINO (without fine-tuning) on the pathological datasets to compare their comprehension of pathologies. While both models have scores of $< 1\%$ AP across all datasets, a qualitative analysis (see Figure A3 in the Appendix H) shows, that the pretrained model seems to grasp the concept of a tissue structure better whereas the former is detecting the entire anatomical structure from background. We also fine-tuned two additional bounding box CL models: one pretrained only on MRI scans from TotalSegmentator, and the other only on CT. The results in Table A6 in the Appendix I indicate that the multi-modal pretraining yields better results (51.7 % AP, Table 1) compared to MRI-only (50.6 % AP) and CT-only (50.9 % AP) pretraining.

**CL Categories:** To evaluate the effect of the number of difficulty categories employed during CL training, we perform a small experiment by training the bounding box CL model with just two difficulty categories, opposed to five difficulty categories used otherwise. The results - Table A7 in Appendix J - indicate that fewer difficulty categories do not increase overall performance: 50.4% AP score compared to 51.7% for bounding box CL in Table 1.

## 4. Conclusion

Our study demonstrates the effectiveness of CL strategies in multimodal medical image object detection. Implementation of bounding box size-based and teacher-guided curricula improved overall detection accuracy, particularly for small and medium-sized objects. However, the lack of improvement on the MSD_Liver dataset necessitates further investigation into dataset-specific factors affecting the effectiveness of CL. We observed that both sorting heuristics can be effectively applied in an anti-curriculum fashion, with only the teacher heuristic being able to match the regular sorting approach. Thus, studying the necessary conditions for successful anti-CL needs further investigation. We also explored pretraining G-DINO on a single modality before fine-tuning on the full pathological dataset. Our results indicate that multi-modal pretraining yields slightly better performance. Despite these promising results, challenges in CL implementation remain. Data-level CL still requires hand-crafted difficulty categories and predefined scheduling. In addition, the computational overhead of CL, especially in the teacher-guided approach, warrants consideration in balancing improved performance with increased training time.

## Acknowledgments

P.V. is funded through the Else Kröner Clinician Scientist Endowed Professorship (reference number: 2022_EKCS.17). A.R. is funded through the Bonfor Startup Postdoc Fellowship (reference number: 2024-1B-10)

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

## Appendix A. Overview of all Pathological Datasets

| Dataset Name | Citation | Mod. | Pathology | Images |
|---|---|---|---|---|
| Yale_BM | (Ramakrishnan et al., 2024) | MRI T1-ce | Brain metastasis | 25,563 |
| BraTS_MET | (Moawad et al., 2023) | MRI T1-ce | Brain metastasis | 30,430 |
| | (Baid et al., 2021) | | | |
| BraTS_GLI | (Menze et al., 2015) | MRI T1-ce | Glioma | 163,066 |
| | (Bakas et al., 2017) | | | |
| MSD_Liver | (Antonelli et al., 2022) | CT-ce | Liver tumor | 19,134 |
| MSD_Hep_Vessel | (Antonelli et al., 2022) | CT-ce | Liver tumor | 13,013 |
| KiTS23_Kidney | (Heller et al., 2023) | CT-ce | Kidney tumor | 32,909 |

Table A1: Overview of pathological datasets used in this work. "-ce" refers to constrast enhancing MRI / CT.

## Appendix B. Training Loss and Validation Scores for all Trained Models

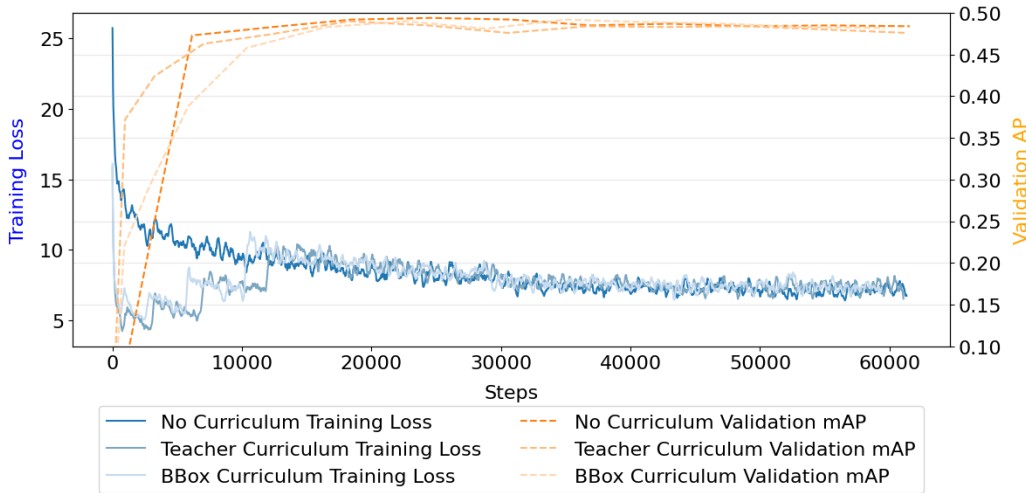

Figure A1: Training loss curves and validation AP plotted for the baseline and the two CL-models. The x-axis denotes training steps and the vertical lines at the top of the graph denote the start of new epochs for each model.

## Appendix C.  Data Distribution for Bounding Box Curriculum

| Datasets / Area Interval | $[0, 0.21)$ | $[0.21, 0.72)$ | $[0.72, 1.75)$ | $[1.75, 3.49)$ | $[3.49, 100]$ | total |
|---|---|---|---|---|---|---|
| Yale_BM | 2037 | 1596 | 881 | 509 | 167 | 5190 |
| BraTS_GLI | 3825 | 5186 | 7933 | 10346 | 10783 | 38073 |
| BraTS_MET | 2969 | 1768 | 1067 | 725 | 227 | 6756 |
| MSD_Liver | 2897 | 1294 | 510 | 196 | 225 | 5122 |
| MSD_Hep_Vessel | 717 | 917 | 758 | 383 | 504 | 3279 |
| KiTS23_Kidney | 1200 | 2619 | 1934 | 1340 | 1477 | 8570 |
| total | 13645 | 13380 | 13083 | 13499 | 13383 | 66990 |

Table A2: Distribution of image slices with ground truth annotations across datasets for the bounding box sorting approach: Based on the smallest bounding box present in a slice, the slice gets sorted into a particular difficulty interval. The intervals are defined by standardized area of the bounding box, with the smallest area intervals being the hardest category.

## Appendix D.  Data Distribution for Teacher Curriculum

| Datasets / AP Interval | $[0, 0.30)$ | $[0.30, 0.69)$ | $[0.69, 0.87)$ | $[0.87, 0.9)$ | $[0.9, 1.0]$ | total |
|---|---|---|---|---|---|---|
| Yale_BM | 878 | 1166 | 1875 | 1069 | 399 | 5387 |
| BraTS_GLI | 6245 | 5242 | 7198 | 10281 | 10437 | 39403 |
| BraTS_MET | 1232 | 1704 | 1993 | 1349 | 698 | 6976 |
| MSD_Liver | 1181 | 2102 | 1337 | 531 | 166 | 5317 |
| MSD_Hep_Vessel | 1024 | 863 | 894 | 509 | 235 | 3525 |
| KiTS23_Kidney | 1676 | 1101 | 2160 | 2288 | 1854 | 9079 |
| total | 12236 | 12178 | 15457 | 16027 | 13789 | 69687 |

Table A3: Distribution of image slices with ground truth annotations and false positive predictions across datasets for the teacher sorting approach: Based on the baseline inference performance, the slice gets sorted into a particular difficulty interval. The intervals are defined by AP scores, with the lowest AP score intervals being the hardest category.

## Appendix E.  Evolution of the Sample Distribution During Training of the Teacher Curriculum Model

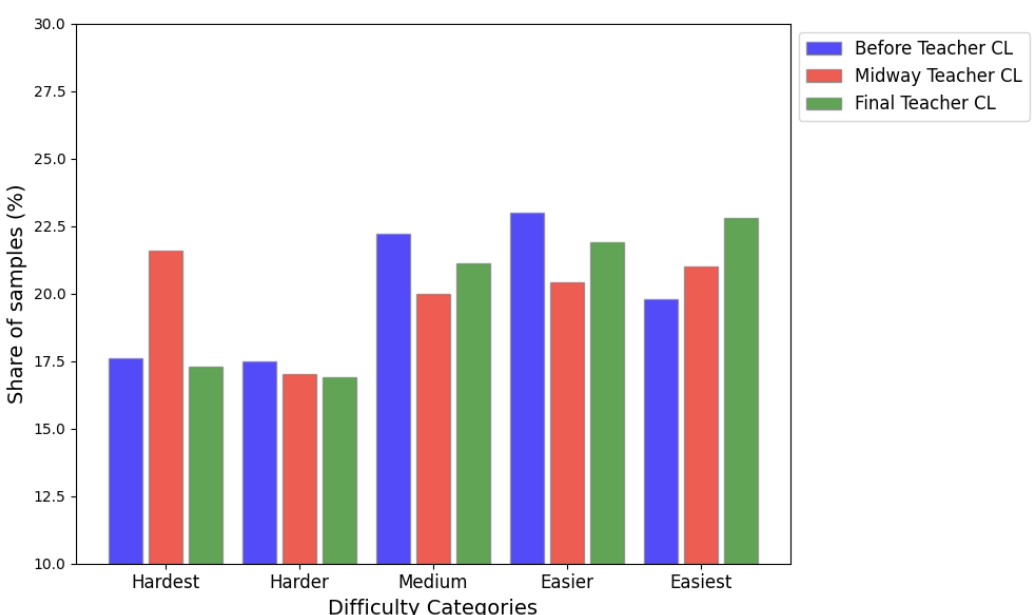

Figure A2: The histogram shows how sample distribution across difficulty categories evolves during Teacher CL trainings. We evaluate the distribution at three stages: (a) before training, (b) midway, after the model has encountered all categories at least once, and (c) after training is complete. As shown in the figure, the midway distribution shifts towards both extremes, reflecting ongoing learning. By the end of training, samples predominantly cluster in the "Easy" category.

## Appendix F.  Evaluation Scores for Anti-Curriculum Approaches

| Dataset | Curr. | AP (%) @0.5 | AP (%) @0.75 | AP (%) | AP (%) large | AP (%) medium | AP (%) small |
|---|---|---|---|---|---|---|---|
| Overall | Without | 69.7 | 50.6 | 46.5 | 69.3 | 62.0 | 30.9 |
|  | Anti Box | 73.9 | 54.2 | 49.5 | 67.6 | 64.0 | 34.1 |
|  | Anti Teacher | **75.5** | **56.2** | **51.2** | **70.1** | **65.1** | **35.8** |
| Yale_BM | Without | 61.5 | 48.2 | 42.2 | - | 69.9 | 37.3 |
|  | Anti Box | **79.2** | **64.3** | **56.0** | - | **81.9** | 51.1 |
|  | Anti Teacher | **79.2** | 63.5 | 55.7 | - | 80.4 | **51.8** |
| BraTS_MET | Without | 66.7 | 54.1 | 45.9 | - | 76.9 | 42.2 |
|  | Anti Box | 82.4 | 65.9 | 57.0 | - | 81.9 | 53.9 |
|  | Anti Teacher | **83.5** | **68.3** | **58.1** | - | **83.9** | **55.2** |
| BraTS_GLI | Without | 84.8 | 72.2 | 65.8 | - | 81.1 | 48.1 |
|  | Anti Box | 85.1 | 72.2 | 65.9 | - | 80.9 | **48.7** |
|  | Anti Teacher | **85.9** | **72.8** | **66.0** | - | **81.2** | 48.6 |
| MSD_Liver | Without | **61.4** | **32.1** | **33.2** | **66.6** | **46.8** | **19.9** |
|  | Anti Box | 55.3 | 25.1 | 28.0 | 66.5 | 41.7 | 15.1 |
|  | Anti Teacher | 57.2 | 28.8 | 30.2 | 65.0 | 42.2 | 18.1 |
| MSD_Hep_Vessel | Without | 69.4 | 37.7 | 39.2 | 62.8 | 43.7 | 15.0 |
|  | Anti Box | **72.0** | **42.8** | 41.5 | 62.7 | **46.6** | 15.7 |
|  | Anti Teacher | 70.4 | 42.3 | **41.6** | **64.8** | 45.9 | **17.5** |
| KiTS23_Kidney | Without | 74.6 | 59.5 | 52.9 | 78.4 | 53.8 | 22.6 |
|  | Anti Box | 69.2 | 54.9 | 48.7 | 73.6 | 50.7 | 20.1 |
|  | Anti Teacher | **76.9** | **61.7** | **55.3** | **80.4** | **57.0** | **23.6** |

Table A4:  Overview of the detection accuracies for the anti-curriculum models on all datasets as well as an overall score (mean over the 6 datasets).

## Appendix G.  Evaluation Scores of Non-Curricula Models Trained on One Modality Only

| Finetuning Modality | Dataset | AP (%) @0.5 | AP (%) @0.75 | AP (%) | AP (%) large | AP (%) medium | AP (%) small |
|---|---|---|---|---|---|---|---|
| CT | Yale_BM | 0.6 | 0.1 | 0.2 | - | 1.1 | 0.1 |
| | BraTS_MET | 0.8 | 0.2 | 0.3 | - | 0.9 | 0.3 |
| | BraTS_GLI | 1.1 | 0.2 | 0.4 | - | 0.8 | 0.2 |
| | MSD_Liver | 55.3 | 24.5 | 27.3 | 65.6 | 39.6 | 14.5 |
| | MSD_Hep_Vessel | 71.2 | 40.2 | 40.5 | 62.4 | 46.5 | 17.6 |
| | KiTS23_Kidney | 77.3 | 61.9 | 54.7 | 79.6 | 55.3 | 23.1 |
| MRI | Yale_BM | 78.6 | 63.8 | 56.1 | - | 79.2 | 52.2 |
| | BraTS_MET | 83.3 | 68.2 | 58.9 | - | 84.4 | 55.7 |
| | BraTS_GLI | 85.1 | 72.4 | 65.8 | - | 81.2 | 48.1 |
| | MSD_Liver | 0.1 | 0.0 | 0.0 | 0.1 | 0.1 | 0.0 |
| | MSD_Hep_Vessel | 0.1 | 0.0 | 0.1 | 0.2 | 0.1 | 0.0 |
| | KiTS23_Kidney | 0.6 | 0.2 | 0.2 | 3.9 | 0.4 | 0.2 |

Table A5: Evaluation on all datasets of two models finetuned on one modality only.

**Finetuning Modality:** The finetuned models shown in Table 1 are trained on a multimodal data from all pathological datasets, making it unclear how each modality contributes individually or whether their combination provides a clear advantage. Table A5 presents test scores for two models finetuned on a single modality without CL. While results are inconclusive, they indicate that combining modalities is not detrimental. As expected, the CT model performs poorly on MRI datasets and vice versa. Moreover, performance on datasets of the same modality as training data is comparable to that of fully finetuned models in Table 1. Specifically, the CT model ranks lowest on MSD_Liver, second lowest on MSD_Hep_Vessel, and second best on KiTS23_Kidney. Meanwhile, the MRI model ranks second best on Yale_BM, ties for second best on BraTS_GLI, and performs best on BraTS_MET, compared to the finetuned models in Table 1.

## Appendix H. Qualitative Comparison of Models Directly After Pretraining on Natural Images vs. Medical Images

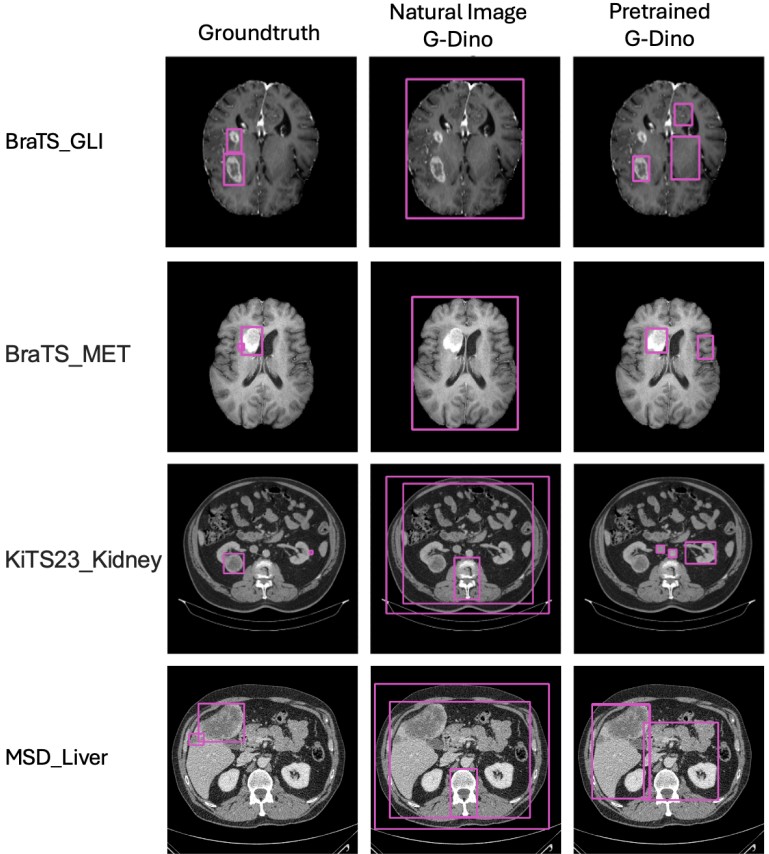

Figure A3: A comparison of the top-three bounding box predictions from the natural image G-DINO model and the G-DINO model pretrained on TotalSegmentator (CT & MRI) across four pathological examples. Neither model was finetuned on the pathological datasets. In instances where fewer than three distinct boxes appear, the same box was predicted multiple times within the top-three. The findings showcase that the natural image G-DINO model typically predicts bounding boxes that encompass the entire region of the human body present in the slice, whereas the medically pretrained G-DINO model sometimes even accurately identifies some tumors or detects the corresponding organ.

In this experiment we test vanilla G-DINO trained on natural image & pretrained G-DINO (on multimodal medical images from TotalSegmentator dataset) directly on the pathological datasets without finetuning to compare their comprehension of pathologies. As expected both models have scores of $\leq 1\%$ AP across all datasets, as they have never been trained on pathological data. However, a qualitative analysis as illustrated in Figure A3, suggests that the pretrained model exhibits a better understanding of tissue structures, whereas the vanilla model struggles to differentiate anatomical features, often detecting the entire image as a foreground object rather than identifying meaningful regions.

## Appendix I. Evaluation Scores for Two Bounding Box Curriculum Models Finetuned After CT-only / MRI-only Pretraining

| Pretraining Modality | Dataset | AP (%) @0.5 | AP (%) @0.75 | AP (%) | AP (%) large | AP (%) medium | AP (%) small |
|---|---|---|---|---|---|---|---|
| CT | Overall | 74.2 | 55.8 | 50.9 | 72.0 | 63.2 | 36.1 |
| | Yale_BM | 77.2 | 63.4 | 56.0 | - | 78.1 | 52.1 |
| | BraTS_MET | 83.3 | 67.1 | 58.0 | - | 83.5 | 54.8 |
| | BraTS_GLI | 85.2 | 73.1 | 66.6 | - | 82.1 | 49.1 |
| | MSD_Liver | 56.8 | 30.9 | 30.7 | 65.1 | 40.0 | 20.0 |
| | MSD_Hep_Vessel | 65.6 | 38.0 | 38.9 | 66.3 | 39.9 | 18.1 |
| | KiTS23_Kidney | 76.9 | 62.0 | 54.9 | 84.7 | 55.8 | 22.7 |
| MRI | Overall | 74.8 | 54.8 | 50.6 | 71.6 | 63.9 | 35.3 |
| | Yale_BM | 78.6 | 64.2 | 56.4 | - | 82.1 | 51.9 |
| | BraTS_MET | 82.2 | 65.0 | 56.9 | - | 83.3 | 53.4 |
| | BraTS_GLI | 85.4 | 72.7 | 66.4 | - | 82.6 | 48.4 |
| | MSD_Liver | 59.4 | 30.5 | 31.9 | 68.0 | 42.2 | 20.7 |
| | MSD_Hep_Vessel | 70.1 | 37.8 | 39.6 | 63.5 | 40.7 | 19.6 |
| | KiTS23_Kidney | 73.1 | 58.7 | 52.3 | 83.2 | 52.7 | 17.7 |

Table A6: Results of Bounding Box CL models for two different pretraining strategies. CT and MRI denote that the models were first pretrained on TotalSegmentator CT / MRI - only and then finetuned using box curriculum.

In this experiment we fine-tuned two additional bounding box CL models: one pretrained only on MRI scans from TotalSegmentator, and the other pretrained only on CT data. The results are tabulated in Table A6. The results indicate that the multi-modal pretraining yields better results (51.7 % AP, Table 1) compared to MRI-only (50.6 % AP) and CT-only (50.9 % AP) pertaining. Moreover, the performance of the multimodal pretrained bounding box CL model is better than the CT-only pretrained bounding box CL model on two out of three CT test datasets, and better than the MRI-only pretrained bounding box CL model on two out of three MRI test datasets.

## Appendix J.  Evaluation Scores of a Bounding Box Curriculum Model Trained With Two Difficulty Categories Only

| Number of CL Categories | Dataset | AP (%) @0.5 | AP (%) @0.75 | AP (%) | AP (%) large | AP (%) medium | AP (%) small |
|---|---|---|---|---|---|---|---|
| 2 | Overall | 74.4 | 55.2 | 50.4 | 66.7 | 64.2 | 35.4 |
|  | Yale_BM | 77.9 | 63.6 | 56.0 | - | 79.2 | 52.4 |
|  | BraTS_MET | 81.9 | 65.4 | 56.6 | - | 83.3 | 53.3 |
|  | BraTS_GLI | 84.6 | 72.1 | 65.8 | - | 81.1 | 48.6 |
|  | MSD_Liver | 60.6 | 31.0 | 32.0 | 61.1 | 44.3 | 21.0 |
|  | MSD_Hep_Vessel | 71.0 | 43.0 | 42.6 | 65.6 | 46.6 | 16.7 |
|  | KiTS23_Kidney | 70.3 | 56.2 | 49.4 | 73.5 | 50.8 | 20.4 |

Table A7: Results table of a bounding box CL model trained using only two difficulty categories.

This ablation study investigates the effect of the number of difficulty categories employed during CL training. For all standard CL-based models depicted in Table 1, the training procedure utilizes five difficulty categories, which are incrementally introduced with each training epoch. After five CL epochs, fine-tuning is then conducted on the entire training set. In contrast, for the ablation, we implemented bounding box CL using only two difficulty categories. Specifically, the model was initially trained on the easier difficulty category for three epochs, after which the second category was introduced and training continued until convergence. The results demonstrate a slight decrease in performance, with an AP score of 50.4% compared to 51.7% AP for the regular bounding box CL (see Table 1).

