# OpenReview forum: "Curriculum Learning for Language-guided, Multi-modal Detection of Various Pathologies"
_MIDL.io/2025/Conference — MIDL 2025 Poster_

### Official Review · Reviewer_Jh4X · 2025-02-17

**Confidence:** 4
**Preliminary Rating:** 4
**Recommendation:** Poster
**Final Rating:** 4

**Summary:**

This paper proposes a method that utilizes G-DINO as the feature extractor, then performs curriculum learning for disease classification tasks.

**Strengths:**

1. Refining the ground truth box approach is interesting and seems to be useful in Figure 2.
2. It is good to follow research trends, i.e., utilizing pre-trained models in medical image analysis tasks.
3. The experiments are thorough. The results seem to be good across different datasets.

**Weaknesses:**

Currently, I vote for 4 for the questions below.
1. Does the pretraining dataset size have an influence on the outcome? From the first part of the pretraining, the G-DINO is pretrained on a large, multi-modal, multi-organ dataset for ROI detection. I would like to see if the pre-training domain impacts the model's performance; e.g., if the model has never been pretrained on a certain modality/task, how well it will perform.
2. Difficulty sorting: It would be kind to have more explanation on how the proposed difficulty sorting approach enhances the model's performance rather than ‘this approach is common in data-level CL and is based on the premise.' Also, I would like to see if there will be a substantial difference if we just divide the samples into two classes: hard and easy, and if these classification results change during model training.

**Detailed Comments:**

Please refer to the weakness part.

**Justification Of The Final Rating:**

Through providing more ablations results, the authors further demonstrate their method's effectiveness, and it is good to see the adaptations of foundational models into the multi-modal medical imaging field. Therefore, I recommend this paper for MIDL.

**Justification Of The Preliminary Rating:**

This paper proposes an interesting way of leveraging pre-trained visual models on medical image tasks, utilizing curriculum learning and refining the ground truth boxes. I would like to see more experiments since it will further demonstrate this method's effectiveness.

**Questions To Address In The Rebuttal:**

Please refer to the weakness part.

---

> ### Author Response · Authors · 2025-03-07
> **Response to Reviewer Jh4X**
>
> Thank you for your encouraging comments and helping us improve the manuscript. As per your suggestions we have modified the manuscript and added results from more experiments. The pointwise reply to all comments and suggestions are listed below in order of your comments.
>
> **Weaknesses**
>
> **W1:** Thank you for your comment and helping us make the experiments more thorough and detailed. To assess the effect of pre-training data we performed two abalation studies
>
> a) Pre-training using only MRI data from TotalSegmentator containing only healthy ROIs.
>
> b) Pre-training using only CT data from TotalSegmentator containing only healthy ROIs.
>
> These models were then fine-tuned with bounding-box curriculum learning on the full pathological dataset (MRI & CT).  The results, detailed in Section 3.6: Ablation Studies: Pretraining and Appendix I, show that the model pre-trained on both modalities performed best on average (51.7% AP) compared to 50.6% AP and 50.9% AP for MRI-only and CT-only pretraining, respectively.
>
>
> **W2:** Thank you for your comment and for helping us make the experiments more thorough and detailed. Regarding the first part of your comment, we have added a more detailed explanation on Curriculum Learning and the intuition behind the difficulty sorting approaches in Section 2.3. Regarding the second part of your comment, we have performed an ablation study by binarizing the difficulty categories into "Easy" and "Hard". Based on these categories we have trained a bounding box CL model and compared it to our original implementation. The results indicate, that binarized difficulty categories underperform compared to five difficulty categories. For more details, please refer to Section 3.6 Ablation Studies: CL Categories and to Appendix J in the updated manuscript. Finally, regarding the third part of your comment, we have visualized the density distribution across categories as the teacher CL model trains. We have evaluated the distribution at three stages (before training, midway of training and at the end of training). By the end of the training, the distribution shifts towards easier categories. Please see Section 3.5 and Appendix E for the visualization of the distribution shift.
>
> **Detailed Comments**
>
> Thank you for your comment. We have replied to each point in the "Weakness" section and modified the manuscript accordingly.
>
> **Questions To Address In The Rebuttal**
>
>  As per your suggestion, we have replied to every comment point-wise in the "Weakness" sections.

---

> > ### Comment · Reviewer_Jh4X · 2025-03-11
> >
> > Dear authors, I have read your responses and your revised manuscript. Your added experiments addressed my concerns and strengthened the paper. As a result, I finalized my final rating for acceptance.

---

> > > ### Author Response · Authors · 2025-03-12
> > > **Response**
> > >
> > > Thank you for acknowledging our efforts and for recommending acceptance!
> > >
> > > Kind regards,
> > >
> > > The Authors!

---

### Official Review · Reviewer_UVRX · 2025-02-19

**Confidence:** 4
**Preliminary Rating:** 4
**Recommendation:** Oral
**Final Rating:** 4

**Summary:**

The authors present a language-guided multi-modal detection framework for multiple pathologies by training a Grounding DINO model. They develop an unified pipeline to generate ground truth bounding boxes from segmentation masks. They also explore two different curriculum learning (CL) strategies - teacher CL and bounding-box CL - to improve detection accuracy across tumor sizes.

**Strengths:**

The paper is well-written, gives a clear overview of the related work and scope of the current work, and presents a comprehensive set of results outlining the effectiveness of their proposed approach.

**Weaknesses:**

1. Some of the newer work in this field seems to be missing in the related works section, e.g. grounded SAM.
2. There was not much discussion on how the text encoder was trained, e.g. what text inputs were used in the pre-training and fine-tuning stages.
3. It would be good to see the AP scores for the vanilla G-DINO and G-DINO after pre-training on TotalSegmentator. This would clearly establish the benefits of both the pre-training and fine-tuning strategies.
4. In section 3.1, instead of describing the individual pre-training and fine-tuning datasets, can we have a table with dataset name, modality, pathology, number of image-mask pairs, etc.?

**Detailed Comments:**

Please see the Weaknesses section.

**Justification Of The Final Rating:**

The changes made to the paper through more discussions and ablation studies look satisfactory and strengthens the paper by better demonstrating the effectiveness of the proposed approach. I recommend acceptance.

**Justification Of The Preliminary Rating:**

The paper reads well and conducts important and novel work in exploring a multi-modal multi-pathology detection framework using G-DINO and CL strategies to improve performance. I believe addressing the previous comments will make it even stronger.

**Questions To Address In The Rebuttal:**

1. More discussion on the language guidance aspect in medical image pre-training and fine-tuning.
2. Inference results with natural image G-DINO and G-DINO after pre-training.

**Special Issue:**

Yes

---

> ### Author Response · Authors · 2025-03-07
> **Response to Reviewer UVRX**
>
> Thank you for your encouraging comments and helping us improve the manuscript. As per your suggestions we have modified the manuscript and added results from more experiments.  The pointwise reply to all comments and suggestions are listed below in order of your comments.
>
> **Weaknesses**
>
> **W1:** Thank you for your comment and for helping us improve our work. We had included works that use Grounded-SAM-like architectures (such as Simtxtseg and Lung grounded-SAM - LuGSAM), but not Grounded-SAM itself. As per your suggestion, we have now updated Section 1: "Introduction and Related Works" (second paragraph).
>
>
> **W2:** Thank you for your comment and pointing this issue out. The text encoder weights are updated during all steps of training (both during pre-training and fine-tuning). The text prompts used in this work are straightforward and simply encompass the class name of the respective target, such as "liver”, "heart”  & "lung_left” during pretraining and "liver_tumor”, "kidney_tumor”  & "brain_metastasis” during fine-tuning. We have updated the manuscript accordingly in Section 2.2 and Section 3.4.
>
>
> **W3:**  As per your suggestings we have now included the results of vanilla G-DINO and pre-trained G-DINO, without fine-tuning on pathology dataset, in the revised manuscript. The results are discussed in Section 3.6: Ablation studies under the heading "Pretraining" and in Appendix H. In summary, both Vanilla G-DINO and pre-trained G-DINO without finetuning had an AP score < 1% across all test sets. This was as per expectation as the pre-training dataset only had bounding boxes from healthy organs while the downstream task focused on pathologies. However as illustrated in Figure A3, when tested on pathology dataset with corresponding prompts, the pre-trained G-DINO can focus on the anatomical structures whereas the vanilla G-DINO detects the complete image as the area-of-interest and separates it from the background. The test scores are not specifically tabulated in the paper, as they are all below 1% AP and thus not meaningful to read. If you wish, we can add them for camera-ready version, though.
>
>
> **W4:** As per your suggestions, we have updated the manuscript. Please see Table A1 in the Appendix A in the updated manuscript for reference. Given the size-limit of 9 pages and the various ablations we were asked to perform, we had to move the table to the appendix.
>
> **Detailed Comments:**
>
> Thank you for your comment. We have replied to each point in the "Weakness" section and modified the manuscript accordingly.
>
> **Questions To Address In The Rebuttal:**
>
> **Q1:** We have updated the revised manuscript as you suggested. Please see the response to point **W2** in the "Weakness" section and Section 2.2 and Section 3.4 in the revised manuscript.
>
> **Q2:** We have updated the revised manuscript based on your suggestion. Please see the response to point **W3** in the "Weakness" section and Section 3.6: Ablation studies and Appendix H in the revised manuscript.

---

> > ### Comment · Reviewer_UVRX · 2025-03-08
> >
> > Thank you for addressing my concerns/suggestions. The revised version looks satisfactory and I recommend acceptance.

---

> > > ### Author Response · Authors · 2025-03-10
> > > **Response**
> > >
> > > Thank you for acknowledging our efforts and for recommending acceptance!
> > >
> > > Kind regards,
> > >
> > > The Authors!

---

> ### Comment · Area_Chair_rdTE · 2025-03-11
>
> please don't forget to update your rating (even if it's the same as before).
> Thank you very much! :)

---

### Official Review · Reviewer_ox4z · 2025-02-21

**Confidence:** 4
**Preliminary Rating:** 2
**Final Rating:** 4

**Summary:**

The paper tackles how to improve multi-modal pathology detection in medical imaging despite challenges such as limited annotations and diverse tumor phenotypes. It asks how a language-guided detection model can be unified across various modalities and anatomical regions while effectively handling varying levels of sample difficulty. The design addresses these questions by converting segmentation datasets into realistic bounding box annotations and integrating two curriculum learning strategies, teacher curriculum and bounding box size curriculum, to progressively train the model from easier to more challenging samples. This approach enhances the model’s generalization and detection accuracy across different pathologies.

**Strengths:**

1. Integrating curriculum learning into a language-guided detection framework improves detection performance across various modalities and pathologies.
2. The novel preprocessing pipeline, which converts segmentation masks into more realistic bounding box annotations, enhances the quality of training data.
3. The curriculum learning strategies are particularly beneficial for detecting small and challenging tumors.

**Weaknesses:**

1. There is no comparison between single-modal performance and multimodal performance, which leaves unclear the contribution of each modality and the motivation for using a bounding box input language model.
2. Several papers illustrate the use of box size as prompts for foundation models [1,2]. The review should clarify the difference and contribution of the proposed box-size curriculum learning design. Furthermore, while the conversion of segmentation masks to bounding boxes using dilation is practical, this preprocessing step is relatively incremental and may not represent a significant novelty compared to standard morphological techniques used in existing literature.
3. The datasets are originally annotated with segmentation masks. It is unclear why this project is designed for a detection task rather than a segmentation task.


[1] Segment anything model for medical image analysis: An experimental study
[2] Leverage weakly annotation to pixel-wise annotation via zero-shot segment anything model for molecular-empowered learning

**Detailed Comments:**

1. What is the performance when the box sizes and samples are ordered randomly?
2. Providing a comparison of the preprocessing pipeline with standard morphological methods for converting segmentation masks to bounding boxes might help the audience understand the benefits of the proposed method.
3. Including a discussion on the computational overhead and training time associated with the curriculum learning approaches compared to the baseline model might help the audience understand the trade-off between performance gains and time costs.

**Justification Of The Final Rating:**

I am satisfied with the additional experimental results and clarifications provided by the author in the rebuttal, which improved the quality of the paper and addressed all my concerns. Therefore, I have increased my rating to accept this paper.

**Justification Of The Preliminary Rating:**

The preliminary rating is justified by the paper’s clear contribution in integrating curriculum learning strategies within a language-guided detection framework. However, concerns regarding the novelty of the preprocessing pipeline relative to standard morphological methods, the lack of comparison between single-modal and multimodal performance, and the limited discussion on computational overhead prevent full endorsement.

**Questions To Address In The Rebuttal:**

Please address the weaknesses and detailed comments in the rebuttal.

---

> ### Author Response · Authors · 2025-03-07
> **Response to Reviewer ox4z**
>
> Thank you for your feedback. We have revised the manuscript as per your suggestions, incorporating additional experiments. Below, we provide a pointwise response to all comments.
>
> **Weaknesses**
>
> **W1**: We appreciate your comments and apologize for the lack of clarity in our motivation. Our algorithm does not use bounding boxes as inputs to the language model; rather, it takes natural language prompts (e.g., “brain_metastasis”) and generates bounding boxes accordingly.
>
> To assess modality-specific fine-tuning, we trained two additional models:
>
> a) Pre-trained on the full TotalSegmentator dataset and fine-tuned only on pathological MRI data.
>
> b) Pre-trained on the full TotalSegmentator dataset and fine-tuned only on pathological CT data.
>
> Both were tested on CT and MRI datasets. The results, detailed in Section 3.6: Ablation Studies: Finetuning Modality and Appendix G, show that single-modality fine-tuning offers no significant advantage over multi-modality fine-tuning. The CT-only model even underperformed on CT datasets compared to the multi-modal baseline, while the MRI-only model performed on par on MRI datasets with the multi-modal bounding box based curriculum learning model.
>
> Additionally, we examined the effect of pre-training modality with two setups:
>
> a) Pre-training using only MRI data from TotalSegmentator containing only healthy ROIs.
>
> b) Pre-training using only CT data from TotalSegmentator containing only healthy ROIs.
>
> These models were then fine-tuned with bounding-box curriculum learning on the full pathological dataset (MRI & CT).  The results, detailed in Section 3.6: Ablation Studies: Pretraining and Appendix I, show that the model pre-trained on both modalities performed best on average (51.7% AP) compared to 50.6% AP and 50.9% AP for MRI-only and CT-only pretraining, respectively.
>
>
>
> **W2**: We apologize for not clearly articulating our objective and novelty. The primary goal of our work is to “predict” bounding boxes using text prompts in a medical imaging context. However, Ref. [1] and [2] use bounding boxes as input prompt to generate a segmentation mask. Our problem statement is different from theirs. Additionally, our proposed algorithm can be used in conjunction with their algorithms to remove the requirement of manual annotation of bounding boxes - which is a tedious process - by replacing it with language prompts. Our contributions include:
>
> (A) Unified Text-Prompt-Based Detection Framework
>
> We propose a text-driven detection model trained across multiple pathologies, body regions, and imaging modalities. Curriculum Learning enhances performance, improving robustness in medical imaging applications.
>
> (B) Novel Preprocessing for Detection Data Generation
>
> Our dilation-based algorithm reduces label noise and overcomes limitations in existing methods, including those in MedSAM. Common approaches, such as tight bounding boxes (Ref. [1]) or slight size adjustments (Ref. [2] and MedSAM), fail with spatially discontinuous lesions. MedSAM even excluded cases where non-connected regions were under 100 pixels. These methods lack a generalizable solution for small, disconnected lesions. In contrast, our approach ensures robust bounding box generation and improved detection across diverse datasets. We have clarified this in Sections 1 and 2.1 of the revised paper.
>
> **W3**: Our decision to experiment on language-based detection rather than segmentation was based on both clinical applicability and research scalability, as detailed below:
>
> (A) Clinical Justification: Detection is often sufficient, eliminating the need for segmentation
>
> • Lesion Counting: Segmentation may misclassify spatially disjoint yet pathologically connected components as separate instances, leading to errors.
> • Lesion Localization: For small tumors, clinicians prioritize identifying presence and location over precise boundaries, as volumetric analysis is not commonly used and can be unreliable for small lesions. Additionally, the small size of segmentation masks relative to the background increases the risk of incorrect predictions.
>
> (B) Scalability Perspective
>
> • Manual annotation for segmentation models (e.g., MedSAM) is tedious. Our method automates detection via text prompts, enabling efficient and reliable bounding box generation for segmentation models.
>
> • Scalable segmentation pipelines can leverage detected ROIs as inputs, eliminating the need for manual bounding boxes
>
>
> **Detailed Comments**
>
> **1** The baseline model trained without CL (mentioned in Table 2 and Figure 3) used randomly ordered box sizes. This is now better clarified in Section 3.4.
>
> **2** Please see our response to Weakness 2
>
> **3** Training and evaluation times (now included in Section 3.5):
>
> • Pre-training (TotalSegmentator): ~4 days
>
> • Fine-tuning (Baseline & CL models): ~2.5 days each
>
> • Teacher CL model: Requires additional fine-tuning of the baseline, which served as the teacher to generate difficulty ratings

---

> > ### Comment · Reviewer_ox4z · 2025-03-09
> > **Response to Rebuttal**
> >
> > I appreciate the author’s efforts during the rebuttal session, which improved the quality of the paper and addressed all my concerns. Thus, I suggest accepting the paper for the MIDL community.

---

> > > ### Author Response · Authors · 2025-03-10
> > > **Response**
> > >
> > > Thank you for acknowledging our efforts and for suggesting acceptance!
> > >
> > > Kind regards,
> > >
> > > The Authors!

---

### Author Rebuttal · Authors · 2025-03-07

**Rebuttal:**

Respected Area Chair and Reviewers:

We would like to thank you for giving us an oppurtunity to revise the manuscript. We have addressed your questions to the best of our ability in the revised manuscript. Newly added text in light of reviewer’s comments has been indicated in magenta color in the revised manuscript for easy following. The major experiments we have performed for this rebuttal are:

A) Perform ablations on pretraining modalities

B) Perform ablations on finetuning modalities

C) Perform ablation on the number of difficulty categories

We have addressed each reviewer’s comments individually below and are happy to clarify any further concerns. Additionally, we have expanded various sections of the manuscript to incorporate your valuable feedback. We look forward to a productive discussion and hope the revised manuscript meets your expectations.

Kind regards,

The Authors

**Supporting Material:**

/attachment/22c497ff8598c9907f27ca69370b2f1967de50ca.pdf

---

### Comment · Area_Chair_rdTE · 2025-03-12
**Additional question for discussion**

First, I would like to thank the authors and reviewers for the productive discussion.

Then, I would like to make a comment that has not been raised in the discussion so far.

If I understand correctly, the proposed model needs two inputs at inference time: an image and a corresponding text prompt that indicates the rough location of the pathology. As such, it seems to me that this model does not perform pathology detection but rather pathology grounding, where we already know that there is a pathology and in which organ, and we now want the model to give us a precise bounding box.

Is this correct? If yes, I would like the authors to clarify this point and replace "pathology detection" to "pathology grounding" (including in the title). If no, can the authors comment on that and maybe run some experiments on healthy subjects to study if the model doesn't predict false positives.

Thanks,
AC rdTE

---

> ### Author Response · Authors · 2025-03-12
> **Response**
>
> Respected Area Chair,
>
> Thank you very much for your question. We are pleased to provide clarification.
>
> During training, for each input image, our network is given all possible classes of the training set as input prompt. Specifically, our training prompt is structured as “glioma . brain metastasis . liver tumor . kidney tumor.”. This means we do not assume a particular organ or pathology beforehand; rather, the network learns these associations during training. Furthermore, we do not assume that every slice contains a pathology. As detailed in Section 3.2 (Data Preprocessing & Training Details) of our manuscript, our training set consists of 199,672 2D slices, of which only 66,990 contain tumors - roughly two-thirds of the slices are pathology-free. This same ratio is maintained in the validation and testing sets.
>
> Based on this setup, we believe that “pathology detection” is indeed the more appropriate term for our work rather than “pathology grounding,” since neither the organ nor the presence of a pathology is assumed a priori. Regarding your suggestion to run experiments on entirely healthy subjects, please note that the false positive rate is already well represented in our reported test scores, given that about two-thirds of the testing set consists of slices without pathologies. However, if you feel that additional experiments on healthy subjects would be beneficial, we are more than willing to conduct them.
>
> We hope this addresses your concerns, and we look forward to any further feedback or questions.
>
> Kind regards,
> The Authors

---

> > ### Comment · Area_Chair_rdTE · 2025-03-14
> >
> > I would like to thank the authors for their clear answer, which addresses my point.
> >
> > I think this ends the discussion for this paper, since all reviewers have finalised their rating.
> >
> > Thanks everyone, I will take all elements into consideration for my recommendation!

---

### Meta-Review · Area_Chair_rdTE · 2025-03-19

**Recommendation:** Accept (Poster)
**Confidence:** 4

**Metareview:**

There is a clear consensus that the rebuttal has strengthened the manuscript and has addressed all the points raised by the reviewers and myself. Having the AC asking questions is a bit unusual, so I'd like to thank the authors for replying to my questions.

First, I would like to point out to the PCs that this work has limited methodological novelty: I don't think we can call the conversion of dilated segmentations to bounding boxes very "novel", and the method combines curriculum learning and G-DINO which have both been proposed before. However, the main interest of this well-written paper is more on the application side, where they combine the aforementioned architecture and training strategy to obtain cross-modal, cross-organ model for pathology detection with a large scope of possibilities. Moreover I think the grounding task tackled here will be of interest to the MIDL community and will foster interesting discussions.

Therefore, following the reviewers' suggestions, I recommend acceptance for this paper.

Finally, I strongly encourage the authors to make a few further modifications to their manuscript:

- I agree with reviewer UVRX that there is not enough detail about how the architecture is implemented and trained. I think the details added after the rebuttal are helping but this is not enough. Please consider further describing this aspect to make your method more reproducible.
- I would like the authors to revise their paper to include their explanation to my question and/or to clarify this aspect in their manuscript.
- Finally, the authors only reported mean scores, and didn't include standard deviations or statistical tests. This point has not been raised by the reviewers, but I think this is a major flaw in the current manuscript and would like the authors to correct for this.